# MAXIMUM ENTROPY COMPETES WITH MAXIMUM LIKELIHOOD

## ABSTRACT

Maximum entropy (MAXENT) method has a large number of applications in theoretical and applied machine learning, since it provides a convenient non-parametric tool for estimating unknown probabilities. The method is a major contribution of statistical physics to probabilistic inference. However, a systematic approach towards its validity limits is currently missing. Here we study MAXENT in a Bayesian decision theory set-up, i.e. assuming that there exists a well-defined prior Dirichlet density for unknown probabilities, and that the average Kullback-Leibler (KL) distance can be employed for deciding on the quality and applicability of various estimators. These allow to evaluate the relevance of various MAXENT constraints, check its general applicability, and compare MAXENT with estimators having various degrees of dependence on the prior, *viz.* the regularized maximum likelihood (ML) and the Bayesian estimators. We show that MAXENT applies in sparse data regimes, but needs specific types of prior information. In particular, MAXENT can outperform the optimally regularized ML provided that there are prior rank correlations between the estimated random quantity and its probabilities.

## 1 INTRODUCTION

The maximum entropy (MAXENT) method was proposed within statistical physics (Jaynes, 1957; Balian, 2007; Pressé et al., 2013), and later on got a wide range of inter-disciplinary applications in data science, probabilistic inference, biological data modeling *etc*; see e.g. (Erickson & Smith, 2013). MAXENT estimates unknown probabilities (that generated data) via maximizing the Boltzmann-Gibbs-Shannon entropy under certain constraints which can be derived from the observed data (Erickson & Smith, 2013). MAXENT leads to *non-parametric* estimators whose form does not depend on the underlying mechanism that generated data (i.e. prior assumptions). Also, MAXENT avoids the zero-probability problem, i.e. when operating on a sparse data—so that certain values of the involved random quantity may not appear due to a small, but non-zero probability—MAXENT still provides a controllable non-zero estimate for this small probability.

MAXENT has has several *formal* justification (Jaynes, 1957; Chakrabarti & Chakrabarty, 2005; Baez et al., 2011; Van Campenhout & Cover, 1981; Topsøe, 1979; Shore & Johnson, 1980; Paris & Vencovská, 1997). But the following open problems are basic for MAXENT, because their insufficient understanding prevents its *valid* applications. *(i)* Which constraints of entropy maximization are to be extracted from data, which is necessarily finite and noisy? *(ii)* When and how these constraints can lead to overfitting, where —due to a *noisy* data—involving more constraints leads to poorer results? *(iii)* How predictions of MAXENT compare with those of other estimators, e.g. the (regularized) maximum likelihood?

Here we approach these open problems via tools of Bayesian decision theory (Cox & Hinkley, 1979). We assume that the data is given as an i.i.d. sample of a *finite* length $M$ from a random quantity with $n$ outcomes and unknown probabilities that are instanced from a non-informative prior Dirichlet density, or a mixture of such densities. Focusing on the sparse data regime $M < n$ we calculate average KL-distances between real probabilities and their estimates, decide on the quality of MAXENT under various constraints, and compare it with the (regularized) maximum-likelihood (ML) estimator. Our main results are that MAXENT does apply to sparse data, but does demand specific prior information. We explored two different scenarios of such information. First, the

unknown probabilities are most probably deterministic. Second, there are prior rank correlations between the inferred random quantity and its probabilities. Moreover, in the latter case the non-parametric MAXENT estimator is better in terms of the average KL-distance than the *optimally* regularized ML (parametric) estimator.

Some of above questions were already studied in literature. (Good, 1970; Christensen, 1985; Zhu et al., 1997; Pandey & Dukkipati, 2013) applied formal principles of statistics (e.g. the Minimum Description Length) to the selection of constraints (question *(i)*). Our approach to studying this question will be direct and unambiguous, since, as shown below, the Bayesian decision theory leads to clear criteria for the validity of MAXENT estimators. We can also compare all predictions with the optimal Bayesian estimator. The latter is normally not available in practice due to insufficient knowledge of prior details, but it still does provide an important theoretical benchmark. Note that (Thomas, 1979; Lebanon & Lafferty, 2002; Kazama & Tsujii, 2005; Altun & Smola, 2006; Dudik, 2007.; Rau, 2011; Campbell, 1999; Friedlander & Gupta, 2005) studied soft constraints that allow incorporation of prior assumptions into the MAXENT estimator making it effectively parametric. Here MAXENT will be taken in its original meaning as providing non-parametric estimators.

This paper is organized as follows. Section 2 recall the tenets of the Bayesian decision theory and describes the data-generation set-up. Section 3 introduces and motivates the Bayesian estimator and the regularized ML estimator. Section 4 recalls the basic formulas of MAXENT, applies them to the studied set-up, and discusses their symmetry features. Section 5 compares predictions of MAXENT with the regularized ML. We close in the last section with discussing open problems. Appendix A shows how to apply MAXENT to categorical data. Appendix B presents our preliminary results on the affine symmetry of MAXENT estimators, and establishes relations with the minimum entropy principle proposed in (Good, 1970; Christensen, 1985; Zhu et al., 1997; Pandey & Dukkipati, 2013).

## 2 BAYESIAN DECISION THEORY

Consider a random quantity $Z$ with values $(z_1, ..., z_n)$ and respective probabilities $q = (q_1, ..., q_n) = (q(z_1), ..., q(z_n))$. We look at an i.i.d. sample of length $M$:

$$\mathcal{D} = (Z_1, ..., Z_M), \qquad m = \{m_k\}_{k=1}^n, \qquad M \equiv \sum_{k=1}^n m_k, \tag{1}$$

where $Z_u \in (z_1, ..., z_n)$ $(u = 1, ..., M)$, and $m_k$ is the number of appearances of $z_k$ in (1). This sample will be an instance of our data, e.g. constraints of MAXENT will be determined from it. The conditional probability of data $\mathcal{D}$ reads

$$P(\mathcal{D}|q_1, ..., q_n) = P(m_1, ..., m_n|q_1, ..., q_n) = M! \prod_{k=1}^n \frac{q_k^{m_k}}{m_k!}. \tag{2}$$

To check the performance of various inference methods, the probabilities $\hat{q}(\mathcal{D}) = \{\hat{q}_k(\mathcal{D})\}_{k=1}^n$ inferred from (1) are compared with true probabilities $q = \{q(z_k)\}_{k=1}^n$ via the KL-distance

$$K[q, \hat{q}(\mathcal{D})] = \sum_{k=1}^n q_k \ln \frac{q_k}{\hat{q}_k(\mathcal{D})}, \tag{3}$$

where concrete forms of $\hat{q}(\mathcal{D})$ are given below. The choice of distance (3) is motivated below, where we recall that it implies the global optimality of the standard (posterior-mean) Bayesian estimator. Another possible choice of distance is the squared (symmetric) Hellinger distance: $\text{dist}_H[q, \hat{q}] \equiv 1 - \sum_{k=1}^n \sqrt{q_k \, \hat{q}_k}$. In our situation, it frequently leads to the same qualitative results as (3).

How to compare various estimators with each other, and decide on the quality of a given estimator? Bayesian decision theory comes to answer this question; see chapter 11 of (Cox & Hinkley, 1979). The theory assumes that the probabilities of $(z_1, ..., z_n)$ are generated from a known probability density $P(q_1, ..., q_n)$ that encapsulates the prior information about the situation. Next it decides on the quality of an estimator $\hat{q}(\mathcal{D})$ via the average distance

$$\langle \overline{K} \rangle = \int \prod_{k=1}^n dq_k \, P(q_1, ..., q_n) \, \overline{K}, \qquad \overline{K} = \sum_{\mathcal{D}} P(\mathcal{D}|q) K[q, \hat{q}(\mathcal{D})]. \tag{4}$$

where $\overline{K}$ is the average of (3) over samples (1) with fixed length $M$. Sometimes the Bayesian decision theory replaces the distance by the utility, loss *etc* (Cox & Hinkley, 1979). Note the difference between the proper Bayesian approach and the Bayesian decision theory; cf. chapters 10 and 11 in (Cox & Hinkley, 1979). The former employs the data for moving from the prior (5) to the posterior (7). It averages over the prior, e.g. when calculating the posterior mean. The latter advises on choosing estimators, whose form may or not may not depend on the prior; see below for examples. The decision theory averages both over the data and over the prior, as seen in (4).

For the prior density of $q = \{q_k\}_{k=1}^n$ we choose the Dirichlet density (or a mixture of such densities as seen below) (Frigyik et al., 2010; Schafer, 1997):

$$P(q_1, ..., q_n \, ; \, \alpha_1, ..., \alpha_n) = \frac{\Gamma[\sum_{k=1}^n \alpha_k]}{\prod_{k=1}^n \Gamma[\alpha_k]} \prod_{k=1}^n q_k^{\alpha_k - 1} \, \delta(\sum_{k=1}^n q_k - 1), \tag{5}$$

where $\Gamma[x] = \int_0^\infty \mathrm{d}y \, y^{x-1} \, e^{-y}$ is Euler's $\Gamma$-function and delta-function $\delta(\sum_{k=1}^n q_k - 1)$ ensures the normalization of probabilities. Parameters $\alpha_k > 0$ determine the prior weight of $q_k$ (Frigyik et al., 2010; Schafer, 1997):

$$\langle q_k \rangle \equiv \int_0^\infty \prod_{l=1}^n \mathrm{d}q_l \, q_k \, P(q_1, ..., q_n \, ; \, \alpha_1, ..., \alpha_n) = \frac{\alpha_k}{A}, \qquad A \equiv \sum_{k=1}^n \alpha_k, \tag{6}$$

where the integration range goes over the simplex $0 \le q_k \le 1$, $\forall k$, and $\sum_{k=1}^n q_k = 1$. Dirichlet density (5) is unique in holding several desired features of non-informative prior density over unknown probabilities; see (Frigyik et al., 2010; Schafer, 1997) for reviews. An important feature of density (5) is that it is conjugate to the multinomial conditional probability (2)

$$P(q_1, ..., q_n | m_1, ..., m_n) = P(q_1, ..., q_n \, ; \, \alpha_1 + m_1, ..., \alpha_n + m_n). \tag{7}$$

Eq. (7) is convenient when studying i.i.d. samples (1) of discrete random quantities. Here we assume that the prior density is known exactly [see however (32)]. In practice, such a knowledge need not be available. For example, it may be known that the prior density belongs to the Dirichlet family, but its hyper-parameters $\{\alpha_k\}_{k=1}^n$ are unknown and should be determined from the data, e.g. via empirical Bayes procedures; see (Frigyik et al., 2010; Schafer, 1997; Claesen & De Moor, 2015; Ran & Hu, 2017; Bergstra & Bengio, 2012) for reviews on hyper-parameter estimation.

## 3 BAYESIAN AND REGULARIZED MAXIMUM LIKELIHOOD (ML) ESTIMATORS

Starting from (4), we find the best estimator in terms of the minimal, average KL-distance:

$$\min[\langle \overline{K} \rangle] = \sum_{\mathcal{D}} P(\mathcal{D}) \min \left[ \int \prod_{k=1}^n \mathrm{d}q_k \, P(q|\mathcal{D}) K[q, \hat{q}(\mathcal{D})] \right], \tag{8}$$

where the minimization goes over inferred probabilities $\{\hat{q}(\mathcal{D})\}$, and where $P(q|\mathcal{D})$ is recovered from $P(\mathcal{D}|q)$: $P(\mathcal{D})P(q|\mathcal{D}) = P(\mathcal{D}|q)P(q)$; cf. (1, 2). The equality in (8) follows from the fact that if $\hat{q}(\mathcal{D})$ minimizes $\int \prod_{k=1}^n \mathrm{d}q_k \, P(q|\mathcal{D}) K[q, \hat{q}(\mathcal{D})]$, then it will minimize each term of the sum for every $\mathcal{D}$, and thus will minimize the whole sum. Then implementing the constraint $\sum_{k=1}^n \hat{q}_k(\mathcal{D}) = 1$ via a Lagrange multiplier, we get from (8):

$$\mathrm{argmin} \left[ \int \prod_{k=1}^n \mathrm{d}q_k \, P(q|\mathcal{D}) \, K[q, \hat{q}(\mathcal{D})] \right] = \left\{ \int \prod_{k=1}^n \mathrm{d}q_k \, q_l \, P(q|\mathcal{D}) \right\}_{l=1}^n. \tag{9}$$

We got in (9) the posterior average, because we employed the KL distance $K[q, \hat{q}(\mathcal{D})]$. The optimal estimator will be different upon using another distance, e.g. KL distance $K[\hat{q}(\mathcal{D}), q]$ of $\hat{q}(\mathcal{D})$ from $q$, or the Hellinger distance. Note that in the proper Bayesian approach the posterior mean is simply postulated to be an estimator, since it is just a characteristics of the posterior distribution. In the present Bayesian decision approach the posterior emerges from minimizing a specific (*viz.* KL) distance. If another distance is used, the posterior mean is not anymore optimal.

If the prior is a single Dirichlet density (5) we get from (7, 9) for the Bayesian estimator:

$$p(z_k) = \frac{m_k + \alpha_k}{M + A}. \tag{10}$$

The average KL-distance (4) for the estimator (10) reads from (7, 2) (denoting $\psi[x] \equiv \frac{\mathrm{d}}{\mathrm{d}x} \ln \Gamma[x]$):

$$\overline{\langle K[q, p] \rangle} = \frac{1}{A} \sum_{k=1}^{n} \alpha_k \psi(1 + \alpha_k) - \psi(1 + A) + \ln(M + A)$$

$$- \frac{\Gamma[M + 1]\,\Gamma[A]}{\Gamma[M + A + 1]} \sum_{k=1}^{n} \sum_{m=0}^{M} \frac{\Gamma[m + 1 + \alpha_k]\,\Gamma[M - m + A + \alpha_k]\,\ln(m + \alpha_k)}{\Gamma[\alpha_k]\,\Gamma[A - \alpha_k]\,\Gamma[m + 1]\,\Gamma[M - m + 1]}. \quad (11)$$

If the prior density is given by mixture of Dirichlet densities with weights $\{\pi_a\}_{a=1}^{L}$:

$$\sum_{a=1}^{L} \pi_a P(q_1, ..., q_n\,;\, \alpha_1^{[a]}, ..., \alpha_n^{[a]}), \qquad \sum_{a=1}^{L} \pi_a = 1, \quad (12)$$

then instead of (6) and (10) we have from (9)

$$\langle q_k \rangle = \sum_{a=1}^{L} \pi_a \frac{\alpha_k^{[a]}}{A^{[a]}}, \qquad A^{[a]} \equiv \sum_{k=1}^{n} \alpha_k^{[a]}, \quad (13)$$

$$p(z_k) = \frac{\sum_{a=1}^{L} \pi_a\,\Phi^{[a]}\,\frac{m_k + \alpha_k^{[a]}}{M + A^{[a]}}}{\sum_{a=1}^{L} \pi_a\,\Phi^{[a]}}, \qquad \Phi^{[a]} \equiv \frac{\Gamma[A^{[a]}]}{\Gamma[M + A^{[a]}]} \prod_{k=1}^{n} \frac{\Gamma[m_k + \alpha_k^{[a]}]}{\Gamma[\alpha_k^{[a]}]}. \quad (14)$$

For a mixture prior density, the Bayesian estimator (14) depends on *all* numbers $\{m_k; \alpha_k^{[1]}, ..., \alpha_k^{[L]}\}$ not just on $m_k$. Below we illustrate that not knowing precisely details of the prior mixture can lead to serious losses when applying Bayesian estimators.

It is interesting (both conceptually and practically) to have a simple estimator, where the dependence on the prior is reduced to a single parameter. A good candidate is the regularized maximum likelihood (ML) estimator (see (Hausser & Strimmer, 2009) for a review):

$$p_{\mathrm{ML}}(z_k) \equiv \frac{m_k + b}{M + nb} = \lambda \frac{m_k}{M} + (1 - \lambda)\frac{1}{n}, \quad \lambda = \frac{M}{M + nb}, \qquad b \geq 0, \quad 0 < \lambda < 1, \quad (15)$$

where the regularizer $b$ (or $\lambda$) takes care of the fact that for a finite sample (1) not all values $z_k$ had a chance to appear (i.e. $m_k = 0$ for them). Then (15) avoids to claim a zero probability due to $b > 0$. Eq. (15) is a shrinkage estimator, where the proper ML estimator $\frac{m_k}{M}$ is shrunk towards uniform distribution $\frac{1}{n}$ by the shrinkage factor $\lambda$. The proper ML estimator $p_{\mathrm{ML}}(z_k)|_{b=0}$ will be shown to be a meaningless estimator for not very long samples (1) producing results that are worse than $\{q(z_k) = \frac{1}{n}\}_{k=1}^{n}$. Moreover, for such samples the correct choice of $b$ (based on the prior information) is crucial, i.e. (15) is generally a parametric estimator. The estimator (15) recovers true probabilities for $M \to \infty$ (Cox & Hinkley, 1979), where $n$ and $b$ are fixed, hence $\lambda \to 1$ in (15).

For the optimal estimator (15), the value of $b$ is found by minimizing the average KL-distance (4). When the prior is given by a Dirichlet density (5), the average KL-distance amounts to (11), where we need to replace $\ln(M + A) \to \ln(M + nb)$ and $\ln(m + \alpha_k) \to \ln(m + b)$. Now (9, 10) imply that for a homogeneous Dirichlet prior, i.e. for (5) with $\alpha_k = \alpha$, we have $b_{\mathrm{opt}} = \alpha$ for the optimal value of $b$, i.e. the regularized ML estimator coincides with the Bayesian estimator: $p_{\mathrm{ML}}(z_k) = p(z_k)$. This does not anymore hold for the mixture of Dirichlet prior densities.

## 4 THE MAXIMUM ENTROPY (MAXENT) METHOD

MAXENT infers probabilities from maximizing the Boltzmann-Gibbs-Shannon entropy

$$S[q] = -\sum_{k=1}^{n} q(z_k) \ln q(z_k), \quad (16)$$

under constraints taken from the sample (1). The rationale of maximizing (16) is that a larger $S$ means a smaller bias (or information) according to several axiomatic schemes (Jaynes, 1957; Chakrabarti & Chakrabarty, 2005; Baez et al., 2011; Van Campenhout & Cover, 1981; Topsøe,

1979; Shore & Johnson, 1980; Paris & Vencovská, 1997; Balian, 2007; Pressé et al., 2013). Note that physical applications of MAXENT operate with constraints that are known precisely, e.g. the mean energy constraint is deduced from the corresponding conservation law (Jaynes, 1957; Balian, 2007; Pressé et al., 2013). Such situations are rare in statistics and machine learning. Hence we need to understand which constraints are to be taken from the noisy data.

First we can apply no constraint and maximize the entropy:

$$q^{[0]}(z_k) = 1/n. \tag{17}$$

The calculation of the average distance is straightforward from (4, 11, 17) both for a single Dirichlet prior and a mixture of such priors. We examplify the single Dirichlet case (5):

$$\overline{\langle K[q, q^{[0]}]\rangle} = \sum_{k=1}^{n} \langle q_k \ln q_k \rangle + \ln n = \frac{1}{A} \sum_{k=1}^{n} \alpha_k \psi(1 + \alpha_k) - \psi(1 + A) + \ln n. \tag{18}$$

Now $\overline{\langle K[q, q^{[0]}]\rangle}$ plays an important role: once (17) is completely data-independent and simply reproduced the prior expectation on the unbiased probabilities, estimators that provide the average KL-distance larger than (18) are meaningless; see below for examples.

Next, we employ the empiric mean of (1) as a constraint for the expected value of $Z$:

$$\mu_1 = \frac{1}{M} \sum_{u=1}^{M} Z_u = \frac{1}{M} \sum_{k=1}^{n} z_k m_k = \sum_{k=1}^{n} q_k z_k. \tag{19}$$

Maximizing (16) under constraint (19) via the Lagrange method leads to the famous Gibbs formula (Jaynes, 1957; Balian, 2007; Pressé et al., 2013):

$$q^{[1]}(z_k) = \frac{e^{-\beta z_k}}{\sum_{l=1}^{n} e^{-\beta z_l}}, \tag{20}$$

where the Lagrange multiplier $\beta$ is found from (19). Appendix presents an example of applying (20) to real data. We order the values of $Z$ as $z_1 < ... < z_n$, and note a specific feature of (20): depending on the sign of $\beta$, we either get

$$q^{[1]}(z_1) \leq ... \leq q^{[1]}(z_n) \quad \text{or} \quad q^{[1]}(z_1) \geq ... \geq q^{[1]}(z_n). \tag{21}$$

One can try to acquire further information from sample (1) by looking at the second empiric moment:

$$\mu_2 = \frac{1}{M} \sum_{u=1}^{M} Z_u^2 = \frac{1}{M} \sum_{k=1}^{n} z_k^2 m_k = \sum_{k=1}^{n} q_k z_k^2. \tag{22}$$

Now we maximize (16) under two constraints (19) and (22):

$$q^{[1+2]}(z_k) = \frac{e^{-\beta_1 z_k - \beta_2 z_k^2}}{\sum_{l=1}^{n} e^{-\beta_1 z_l - \beta_2 z_l^2}}, \tag{23}$$

where Lagrange multipliers $\beta_1$ and $\beta_2$ are found from solving both (19) and (22). Eqs. (20, 23) make obvious how to involve other (fractional) moments. The maximizations of (16) lead to unique results, because (16) is a concave function of $\{p_k\}_{k=1}^{n}$, while the moment constraints are linear.

Let the values $(z_1, ..., z_n)$ of $Z$ be subject to affine transformation

$$\widetilde{z}_k = \mathcal{F}(z_k), \quad \mathcal{F}(z) = gz + h, \quad k = 1, ..., n. \tag{24}$$

Hence as a result of transformation (24): $\mu_1 \to \widetilde{\mu}_1 = g\mu_1 + h$ and $\mu_2 \to \widetilde{\mu}_2 = g^2\mu_2 + h^2 + 2gh\mu_1$; see (19, 22). These relations show that (24) leaves the inferred probabilities (20, 23) invariant, because the resulting set of equations for the unknowns in (20, 23) are identical for both the original $(z_1, ..., z_n)$ and transformed values (24). Likewise, involving first $p$ moments $\mu_1, ..., \mu_p$ produces affine-invariant probabilities. Note that involving only (22) [without involving (19)] will lead to the invariance of the probabilities with respect to a limited affine-symmetry, where $h = 0$ in (24). Another example of limited affine symmetry is involving the fractional moment $\sum_{k=1}^{n} q_k \sqrt{z_k}$ (for $z_k \geq 0$ and instead of (19, 22)). Then the probabilities $q^{[1/2]}(z_k) \propto e^{-\beta_{1/2}\sqrt{z_k}}$ will stay intact only

under $h = 0$ and $g > 0$ in (24). Note in this context that the ML estimator (15) is invariant with respect (24) with an arbitrary bijective $\mathcal{F}$, which keeps the values of $\widetilde{z}_k$ different.

The symmetry features of various estimators are clearly important, though we so far have no analytical results that would relate them to the estimation quality quantified by (4). But we noted from numerical comparison of MAXENT estimators based on various constraints, that estimators with the largest affine symmetry—i.e. (24) with arbitrary $g$ and $h$—tend to be better in terms of the average KL-distance (4). Intuitively, higher (affine) symmetry should be related to higher susceptibility with respect to noises; see Appendix B for further results.

## 5 NUMERICAL RESULTS

### 5.1 A SINGLE DIRICHLET DENSITY

Recall that maximization of entropy (16) can be applied if there is no prior information that distinguishes one probability from another. If such information is present, MAXENT is generalized to the minimum relative entropy method (Shore & Johnson, 1980). We shall not study this generalization here. Hence to ensure applicability of MAXENT, we always choose prior densities such that $\langle q_k \rangle = \frac{1}{n}$; i.e. all $n$ values are equally likely to be generated, on average. As seen from (6), for a single prior Dirichlet density (5) condition $\langle q_k \rangle = \frac{1}{n}$ implies:

$$\alpha_k = \alpha, \qquad k = 1, ..., n, \qquad (25)$$

Now recall from (15, 10) that under (25) the Bayesian and the regularized ML coincide. I.e. we conclude that the regularized ML is a better estimator than MAXENT (under any constraint).

Though $\langle q_k \rangle = \frac{1}{n}$ does not depend on $\alpha$, the most probable values $\widetilde{q}_k$ of $q_k$ do depend on the magnitude of $\alpha$. Finding $\widetilde{q}_k$ from (5, 25) amounts to maximizing $\mathcal{L}(q) = (\alpha - 1) \sum_{k=1}^{n} \ln q_k + \gamma \sum_{k=1}^{n} q_k$, where the Lagrange multiplier $\gamma$ ensures $\sum_{k=1}^{n} q_k = 1$. For $\alpha > 1$, $\mathcal{L}(q)$ is a concave function of $q$, and its global maximum is found after differentiating it. Hence $\widetilde{q}_k$ holds

$$\widetilde{q}_k = 1/n \quad \text{for} \quad \alpha > 1, \quad k = 1, ..., n. \qquad (26)$$

For $\alpha < 1$, $\mathcal{L}(q)$ is a convex function, it does not have local maxima with $q_k > 0$ ($k = 1, ..., n$). Its maxima are located at points, where $q_k = 0$ for certain $k$. Repeating this argument, we see that the maxima of $\mathcal{L}(q)$ are at those points, where a possibly large number of $q_k$ are zero:

$$\widetilde{q}_k = 0 \quad \text{or} \quad \widetilde{q}_k = 1, \quad \text{for} \quad \alpha < 1, \quad k = 1, ..., n, \qquad (27)$$

which means deterministic probabilities. Eq. (27) is consistent with $\langle q_k \rangle = 1/n$, because there are $n$ equivalent most probable values.

Let us start with the regime $\alpha > 1$; cf. (26). Table 1 compares predictions of (15) with those of MAXENT solutions (20) and (23) for the Dirichlet prior (5) holding (25) with $\alpha = 2$. It is seen that MAXENT is meaningless, because the trivial estimator (17) provides a smaller average KL-distance; cf. (18). For the Bayesian estimator even $M = 1$ leads to a meaningfull prediction; e.g. for parameters of Table 1 we have: $\langle \overline{K}_{\text{Bayes}} \rangle |_{M=1} = 0.224 < 0.225$.

The above conclusion holds more generally (as we checked numerically): for the homogeneous Dirichlet prior (25) with $\alpha \geq 1$, MAXENT estimators (20, 19) and (23, 19, 22) are meaningless at least in the sparse data regime $M < n$. This puts a serious limitation on the validity of MAXENT.

The situation changes for sufficiently small values of $\alpha$ in the regime (27); see Table 2 for $\alpha = 0.1$. Here the MAXENT estimators are meaningful provided that the sample length $M$ is sufficiently large (but still in the sparse data regime $M < n$): (20, 19) is meaningful for $M \geq 9$ ($M < n = 60$), while the estimator (23, 19, 22) is meaningful for $M \geq 25$; see Table 2. Though predictions of MAXENT are still far from those of the Bayesian estimator, we should recall that the latter estimator is parametric, i.e. it depends on the prior (via the parameter $\alpha$) in contrast to MAXENT estimators. Table 2 demonstrates the overfitting phenomenon: for $9 \leq M \leq 15$ the MAXENT estimator (20, 19) is meaningful, but adding the second constraint makes the MAXENT estimator (23, 19, 22) not meaningful. The situation is worsened since (22) is again estimated from the noisy data and gathers more noise than information. This overfitting disappears for larger values of $M$, i.e. $M \geq 25$, as Table 2 demonstrates. Now adding the second constraint (22) is beneficial.

Table 1: For $n = 60$ and $z_k = k$ ($k = 1, ..., n$) we show the average KL-distance (4) for various estimators. The full affine symmetry (24) holds for all shown probabilities. $M$ is the length of sample (1). The initial prior Dirichlet density (5) holds (25) with $\alpha_k = 2$. Eq. (18) equals $\langle \overline{K[q, q^{[0]}]} \rangle = 0.225$, i.e. values of the average KL-distance larger than $0.225$ are *meaningless*. $\langle \overline{K}_{\text{Bayes}} \rangle$ is the averaged KL-distance for the Bayes estimator (10) that for this case coincides with the optimally regularized ML estimator. $\langle \overline{K}_1 \rangle$ and $\langle \overline{K}_{1+2} \rangle$ are defined (resp.) via (20, 19) and (23, 22). The averages are found numerically (applies to all Tables): first we generate $10^2$ instances of $\{q_k\}_{k=1}^n$ from the Dirichlet density, and then for each instance we generate $10^2$ samples (1). Such parameters lead to 3-digit precision, as reported.

| $M$ | $\langle \overline{K}_{\text{Bayes}} \rangle$ | $\langle \overline{K}_1 \rangle$ | $\langle \overline{K}_{1+2} \rangle$ |
|---|---|---|---|
| 35 | 0.177 | 0.236 | 0.247 |
| 25 | 0.188 | 0.240 | 0.260 |
| 15 | 0.202 | 0.259 | 0.301 |

Table 2: The same as in Table 1, but for $\alpha_k = \alpha = 0.1$ in (25). Eq. (18) gives $\langle \overline{K[q, q^{[0]}]} \rangle = 1.798$, i.e. values of the average KL-distance larger than $1.798$ are meaningless.

| $M$ | $\langle \overline{K}_{\text{Bayes}} \rangle$ | $\langle \overline{K}_1 \rangle$ | $\langle \overline{K}_{1+2} \rangle$ |
|---|---|---|---|
| 55 | 0.233 | 1.756 | 1.685 |
| 45 | 0.276 | 1.700 | 1.643 |
| 35 | 0.338 | 1.723 | 1.680 |
| 25 | 0.428 | 1.753 | 1.717 |
| 15 | 0.606 | 1.770 | 2.164 |
| 11 | 0.730 | 1.762 | 4.946 |
| 9 | 0.818 | 1.774 | 11.63 |
| 7 | 0.916 | 1.848 | 32.24 |

Table 3: The same as in Table 1, but the initial prior density is a Dirichlet mixture given by (12, 30) with $\alpha_0 = 0.3$ and $\epsilon = 1.1$. The average KL-distance $\langle \overline{K[q, q^{[0]}]} \rangle$ for the trivial estimator (17) equals $0.212$, i.e. values of the average KL-distance larger than $0.212$ are *meaningless*; cf. (18). $\langle \overline{K}_{\text{Bayes}} \rangle$ and $\langle \overline{\mathbf{K}}_{\text{Bayes}} \rangle$ refer to (14) and (32), respectively. $\langle \overline{K}_{\text{ML}} \rangle_{b=b_{\text{opt}}}$ and $\langle \overline{K}_{\text{ML}} \rangle_{b=1}$ refer to regularized ML estimator (15) under $b = 1$ and the optimal value of $b$ found from numerically minimizing $\langle \overline{K}_{\text{ML}} \rangle$. The optimal value $b_{\text{opt}}$ of $b$ changes from $2.46$ for $M = 35$ to $2.65$ for $M = 1$. We also report the value of $\langle \overline{K}_{\text{ML}} \rangle_{b=1}$ with a sensible value of $b$ to confirm that if $b$ is not chosen properly, then the corresponding (regularized) ML estimator (15) is meaningless. $\langle \overline{K}_1 \rangle$ is defined via (4, 20). $\langle \overline{K}_{1+2} \rangle$ is not shown, since $\langle \overline{K}_{1+2} \rangle > \langle \overline{K}_1 \rangle$ for $35 \geq M \geq 1$. We do not show $\langle \overline{K}_1 \rangle |_{M=1}$, since it is larger than the average KL-distance for all other estimators.

| $M$ | $\langle \overline{K}_{\text{Bayes}} \rangle$ | $\langle \overline{\mathbf{K}}_{\text{Bayes}} \rangle$ | $\langle \overline{K}_{\text{ML}} \rangle_{b=b_{\text{opt}}}$ | $\langle \overline{K}_{\text{ML}} \rangle_{b=1}$ | $\langle \overline{K}_1 \rangle$ |
|---|---|---|---|---|---|
| 35 | 0.014 | 0.206 | 0.180 | 0.204 | 0.048 |
| 25 | 0.015 | 0.207 | 0.188 | 0.210 | 0.053 |
| 15 | 0.017 | 0.209 | 0.197 | 0.214 | 0.065 |
| 11 | 0.022 | 0.209 | 0.201 | 0.215 | 0.077 |
| 7 | 0.035 | 0.209 | 0.205 | 0.215 | 0.105 |
| 5 | 0.052 | 0.210 | 0.207 | 0.214 | 0.141 |
| 3 | 0.083 | 0.210 | 0.209 | 0.213 | 0.268 |
| 1 | 0.150 | 0.211 | 0.211 | 0.212 | — |

Table 4: The same as in Table 3, but for different values of $M$. Here $\langle \overline{K}_{1+2} \rangle$ refers to MAXENT estimator (23) with constraints (19, 22). The Bayesian estimator is found from (14, 30). For this range of sufficiently large $M$ the MAXENT estimator (23) performs better than (23): $\langle \overline{K}_{1+2} \rangle < \langle \overline{K}_1 \rangle < \langle \overline{K}_{\mathrm{ML}} \rangle_{b=b_{\mathrm{opt}}}$.

| $M$ | $\langle \overline{K}_{\mathrm{Bayes}} \rangle$ | $\langle \overline{K}_{\mathrm{ML}} \rangle_{b=b_{\mathrm{opt}}}$ | $\langle \overline{K}_{\mathrm{ML}} \rangle_{b=1}$ | $\langle \overline{K}_1 \rangle$ | $\langle \overline{K}_{1+2} \rangle$ |
|-----|------|-------|-------|-------|-------|
| 45 | 0.015 | 0.172 | 0.196 | 0.045 | 0.042 |
| 65 | 0.014 | 0.157 | 0.180 | 0.042 | 0.035 |
| 85 | 0.014 | 0.145 | 0.164 | 0.040 | 0.031 |
| 241 | 0.013 | 0.087 | 0.091 | 0.038 | 0.024 |

## 5.2 MIXTURE OF DIRICHLET DENSITIES

For modeling more complex types of prior information about the unknown probabilities $\{q_k\}_{k=1}^n$, we shall assume that the prior density is a mixture of two Dirichlet densities; see (12). Relations $\langle q_k \rangle = \frac{1}{n}$ ($k = 1, ..., n$) will be still kept, since they are necessary for applying MAXENT. Now we assume that that there are (prior) *conditional* rank correlation between the values $(z_1, ..., z_n)$ of $Z$—ordered as $(z_1 < ... < z_n)$—and its probabilities $(q_1, ..., q_n)$. For one component of the mixture, the probabilities $(q_1, ..., q_n)$ prefer to be ordered as in $(q_1 < ... < q_n)$. For another component they tend to be ordered in the opposite way $(q_1 > ... > q_n)$. This type of prior knowledge can be modeled via a mixture (12) of two Dirichlet priors with $L = 2$, $\pi_1 = \pi_2 = \frac{1}{2}$, and

$$\alpha_1^{[1]} < ... < \alpha_n^{[1]}, \qquad \alpha_1^{[2]} > ... > \alpha_n^{[2]}, \tag{28}$$

$$\frac{\alpha_k^{[1]} - \alpha_l^{[1]}}{A^{[1]}} = \frac{\alpha_l^{[2]} - \alpha_k^{[2]}}{A^{[2]}}, \quad \text{for any} \quad k, l = 1, ..., n, \tag{29}$$

where (29) ensures the needed $\langle q_k \rangle = \frac{1}{n}$, as seen from (13). A simple case that leads to (28, 29) is

$$L = 2, \quad \pi_1 = \pi_2 = \frac{1}{2}, \quad \alpha_k^{[1]} = \alpha_0 + \epsilon(k-1), \quad \alpha_k^{[2]} = \alpha_0 + \epsilon(n-k), \qquad k = 1, ..., n, \tag{30}$$

where $A^{[1]} = A^{[2]} = \alpha_0 n + \frac{\epsilon n(n-1)}{2}$. Recall that for a mixture of Dirichlet densities the Bayes estimator (14) and the optimally regularized ML estimator (15) are different.

For numerical illustration we choose $\{z_k = k\}_{k=1}^n$. Prior probability densities generated via (30) will be employed with $z_1 < ... < z_n$. Now Tables 3 and 4 show that for $M \geq 5$ the MAXENT estimator (20, 19) is clearly better than the *optimally* regularized ML estimator (15):

$$\langle \overline{K}_1 \rangle < \langle \overline{K}_{\mathrm{ML}} \rangle_{b=b_{\mathrm{opt}}}, \tag{31}$$

where the optimal value of $b$ is found from minimizing the averaged KL-distance (4). Moreover, for $M \geq 7$, we see that $\langle \overline{K}_1 \rangle$ is closer to the optimal $\langle \overline{K}_{\mathrm{Bayes}} \rangle$ than to $\langle \overline{K}_{\mathrm{ML}} \rangle_{b=b_{\mathrm{opt}}}$. Note that such threshold values for $M$ do depend on the assumed prior density and on $n$.

For $M \to \infty$ the performance of the optimally regularized ML estimator (15) (for a fixed $b \sim 1$) will be better than MAXENT with any finite number of constraints, since the regularized ML converges to the true probabilities for $M \to \infty$ (Cox & Hinkley, 1979), while MAXENT does not. But as Table 4 shows, MAXENT with constraints (19) or (19)+(22) still performs better than the optimally regularized ML even for $M$ as large as 241 (for $n = 60$).

Table 3 shows that MAXENT with two constraints (19, 22) performs worse than the method under the single constraint (19) although the affine invariance (24) of probabilities holds. This *overfitting* situation changes for larger values of $M$, i.e. $M \geq 45$, as Table 4 demonstrates.

To stress the relevance of rank correlations, we note that the advantage (31) of MAXENT closely relates to the agreement between (28) and the ordering $(z_1 < ... < z_n)$ of $Z$. If the vector $(z_1 < ... < z_n)$ is randomly permuted, and employed for values of $Z$, predictions of MAXENT become meaningless even for rather large values of $M > n$.

Recall that both the Bayesian (14) and the regularized ML estimator (15) are parametric, i.e. the very their form depends on the prior, which is frequently not available in practice. Hence we need

to understand how strong is dependence. Let us assume that one has to employ a Bayesian estimator without knowing the full form of the equal-weight mixture (30). Instead one knows the average values of $\alpha_k = \frac{1}{2}[\alpha_0 + \epsilon(k-1)] + \frac{1}{2}[\alpha_0 + \epsilon(n-k)]$ from (30), prescribes them to a single Dirichlet prior (5, 25) and builds up from (10) an estimator

$$\mathbf{p}(z_k) = \frac{m_k + \alpha_0 + \epsilon(n-1)/2}{M + n\alpha_0 + \epsilon n(n-1)/2}. \tag{32}$$

The performance of this perturbed Bayesian estimator deteriorates and gets worse than that of the MAXENT solution: $\langle \overline{K}_1 \rangle < \langle \overline{\mathbf{K}}_{\text{Bayes}} \rangle$; see Table 3. Likewise, the choice of $b$ in the regularized ML estimator (15) is important. If just some reasonable value is chosen instead of the optimal one—e.g. $b = 1$ instead of $b \sim 2.5$ in Table 3—then the ML estimator can turn meaningless; see Table 3 for $M \leq 15$. In this context, we emphasize that the Bayes estimator and the optimally regularized ML estimator are never meaningless even for $M = 1$. For parameters of Table 3, the MAXENT estimator (20) becomes meaningless for $M \leq 3$.

## 6 SUMMARY AND DISCUSSION

The maximum entropy (MAXENT) method provides non-parametric estimators for inferring unknown probabilities (Erickson & Smith, 2013). MAXENT is widely applied both in statistical physics and probabilistic inference. However, its physical applications are mostly data-free and are based on additional principles (e.g. conservation laws (Balian, 2007; Pressé et al., 2013)) that are normally absent in statistics and machine learning. Hence we needed a systematic approach towards understanding the validity limits of MAXENT as an inference tool.

Here we presented a Bayesian decision theory approach that allows to determine on whether MAXENT is applicable at all, i.e. whether it is better than a random guess. It also allows to compare different estimators with each other (e.g. to compare MAXENT with the regularized maximum likelihood), and study the relevance of various constraints employed in MAXENT.

Our results are summarized as follows. MAXENT does apply to a sparse data, but demands specific prior information. Here sparse means $M < n$, i.e. the sample length $M$ is smaller than the number of probabilities $n$ to be inferred. We explored two different scenarios of such prior information. First, the unknown probabilities generated by homogeneous Dirichlet density (25) are most probably deterministic. Second, there are prior rank correlations between the random quantity and its probabilities. This seems to be the simplest prior information that makes MAXENT applicable and superior over the optimally regularized maximum-likelihood estimator. Our approach is capable of describing several phenomena that are relevant for applying and understanding estimators: overfitting (i.e. adding more noisy constraints leads to poorer inference), instability of optimal Bayesian parametric estimators with respect to variation of prior details, inapplicability of non-parametric MAXENT estimators to very short samples *etc*.

Several important problems were uncovered by this study and should be addressed in future. First of all, this concerns the applicability of MAXENT to a categorical data, where the values $(z_1, ..., z_n)$ of the random variable $Z$ in sample (1) are not numerical, but instead refer to certain distinguishable categories. The major difference between maximum likelihood and MAXENT estimator is that the former freely applies to categorical data. In contrast, MAXENT does depend on the concrete numerical implementation (i.e. *encoding*) of data, though this dependence is somewhat weakened by the affine symmetry (24). Thus an open problem demands considering various encoding schemes in view of their applicability to MAXENT estimators. (In this paper we in fact assumed the simplest encoding via natural numbers; see Tables.) Appendix A reports preliminary results in this direction along with a real data example. The second open problem relates to the influence of affine symmetries on the performance of various MAXENT estimators. We observed numerically that the constraints which produce affine-invariant probabilities produce better estimators; see after (24). Preliminary results along this direction are given in Appendix B, where we also show relations of our results with the minimum entropy principle proposed in (Good, 1970; Christensen, 1985; Zhu et al., 1997; Pandey & Dukkipati, 2013) for contraint selection.

ACKNOWLEDGMENTS

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

## A    APPENDIX: MAXENT APPLIES TO CATEGORICAL DATA

The MAXENT method can be applied to any multinomial data (1), provided that numeric values $(z_1, ..., z_n)$ of the random quantity $Z$ are given. MAXENT estimators depend on $(z_1, ..., z_n)$ modulo the affine symmetry (24). This creates a problem in applying MAXENT to categorical data, since for MAXENT one now needs a specific encoding of categorical $Z$ into a numeric representation $(z_1, ..., z_n)$ of categories. Recall that there is a degree of arbitrariness in choosing the regularizer in maximum likelihood (ML) estimator, or in choosing prior parameters for Bayesian inference. Here the arbitrariness lies in different encodings. In practice, the proper encoding of categories arises in any problem dealing with categorical data. If categories are ordinal (e.g. military ranks, education levels), then one can use $z_1 < z_2 < ... < z_n$ encoding. However, for nominal categories (e.g. ethnicity, preference, disease) there is no such ordering.

Let us illustrate the MAXENT method with the simple data from pre-election presidential polling conducted in 1988, where out of $M = 1447$ voters $m_1 = 727$ preferred Bush, $m_2 = 583$ preferred Dukakis, and $m_3 = 137$ preferred other candidates or had no preference. The data together with its Bayesian analysis is taken from (Gelman et al., 2013).

Here our random variable $Z$ is voter's preference with three outcomes $(z_1, z_2, z_3)$ = ('Bush', 'Dukakis', 'Other') and unknowns $(q_1, q_2, q_3)$, which just represent the fractions of the population with each preference. The goal here is to estimate $q_1 - q_2$, i.e. whether Bush has more support than Dukakis. One can assume the Dirichlet noninformative prior distribution for $(q_1, q_2, q_3)$ with parameters $\alpha_1 = \alpha_2 = \alpha_3 = 1$, compute the posterior means of $q_1$ and $q_2$ ($\hat{q}_1, \hat{q}_2$) and take the difference (Gelman et al., 2013). The results show that Bush has more support: $\hat{q}_1 > \hat{q}_2$.

Since the data is purely categorical, we shall apply the frequency encoding for MAXENT: each category is represented with its frequency in the data set, e.g. in this example $(z_1, z_2, z_3)$ = $(0.502, 0.403, 0.095)$. Now empiric mean is equal to $0.42$ and the maximizing solutions of (16) with $\sum_{k=1}^{3} q_k z_k = 0.42$ are $(q^{[1]}(z_1), q^{[1]}(z_2), q^{[1]}(z_3))$ = $(0.535, 0.36, 0.105)$. Thus, also the MAXENT result shows more support for Bush.

To see if the prediction of MAXENT is reliable (on average) here, the same Bayesian decision model for these samples is set up, where first a sample of $(q_1, q_2, q_3)$ is drawn from the Dirichlet distribution

with $\alpha_1 = \alpha_2 = \alpha_3 = 1$, and then using this sample as category probabilities, categorical data sets of size $M$ are generated with categories replaced by its frequency encodings. The process is repeated and the average $\langle \overline{K}_1 \rangle$ from (4) is computed via generating $10^3$ instances of $\{q_k\}_{k=1}^3$ and $10^3$ categorical samples. For the present case $\alpha_1 = \alpha_2 = \alpha_3 = 1$ and $n = 3$, we have $\langle \overline{K[q, q^{[0]}]} \rangle = 0.265$ from (18).

Now for $M > 15$ we get that $\langle \overline{K}_1 \rangle < \langle \overline{K[q, q^{[0]}]} \rangle$, i.e. the MAXENT solution is reliable. For example, at $M = 17$ we have $\langle \overline{K}_{\text{Bayes}} \rangle = 0.046 < \langle \overline{K}_1 \rangle = 0.116 < \langle \overline{K[q, q^{[0]}]} \rangle = 0.265$, where $\langle \overline{K}_{\text{Bayes}} \rangle$ refers to the Bayesian (posterior mean) estimator (10). Already for $M = 47$ predictions of MAXENT are close to those of the optimal Bayesian estimator $\langle \overline{K}_{\text{Bayes}} \rangle = 0.019 < \langle \overline{K}_1 \rangle = 0.034$. For the actual sample size $M = 1447$, we get even closer results $\langle \overline{K}_{\text{Bayes}} \rangle = 0.00087 < \langle \overline{K}_1 \rangle = 0.014$.

To summarize the present real categorical data example, we saw that the frequency encoding of the categorical variable allows to apply MAXENT. The MAXENT estimator (19) agrees with Bayesian estimator, and is going to be reliable already for modest sample sizes $M > 15$. For a sufficiently large $M$ the average KL distance of the MAXENT estimator gets close to that of the (optimal) Bayes estimator.

## B   APPENDIX: AFFINE SYMMETRY AND THE MINIMUM ENTROPY PRINCIPLE

Above we focused on MAXENT estimators (20, 19) (the first empiric moment is fixed) or (23, 19, 22) (the first and second empiric moments are fixed). As discussed around (24), both (19) and (22) lead to affine-invariant probabilities. We studied several alternative constraints that do not have the full affine symmetry, i.e. this symmetry is partial and relates to restriction on the parameters in (24). An example of this is constraining the square-root (fractional) moment [cf. (20, 23)]

$$q^{[1/2]}(z_k) = \frac{e^{-\beta_{1/2}\sqrt{z_k}}}{\sum_{l=1}^n e^{-\beta_{1/2}\sqrt{z_l}}}, \tag{33}$$

$$\sum_{k=1}^n q_k \sqrt{z_k} = \frac{1}{M} \sum_{u=1}^M \sqrt{Z_u}, \tag{34}$$

where $\beta_{1/2}$ is determined from (34), and where we assumed $z_k > 0$. For estimator (33) the symmetry (24) is kept under $g > 0$ and $h = 0$. We denote the corresponding average KL distances by $\langle \overline{K}_{1/2} \rangle$.

Let us now compare two different MAXENT estimators each one employing its own constraint; e.g. we compare (20) with (33). We saw from extensive numeric simulations that whenever these constraints have different degrees of the affine symmetry, then the estimator having the largest symmetry provides a smaller average KL distance. A particular example of this general relation is:

$$\langle \overline{K}_1 \rangle < \langle \overline{K}_{1/2} \rangle, \tag{35}$$

which was verified on parameters of Tables 1–4.

Recall that Refs. (Good, 1970; Christensen, 1985; Zhu et al., 1997) proposed the minimum entropy principle: when comparing two possible contraints to be employed in the maximum entropy method, then it is preferable to use the one that provides the smaller (maximized) entropy. The heuristic motivation of the principle is that it avoids overfitting by not insisting too much on the entropy maximization. This principle was motivated via the minimum description length in (Pandey & Dukkipati, 2013).

We ask whether in cases similar to (35) we can compare the average entropies, i.e. for (35) we compare $\langle \overline{S[q^{[1]}(z_k)]} \rangle$ and $\langle \overline{S[q^{[1/2]}(z_k)]} \rangle$, where the averages are defined as in (4). In all cases we were able to check, relations similar to (35) are accompanied by the result that the constraint which provide a smaller average KL distance (i.e. a better costraint) also has a smaller average entropy, e.g.

$$\langle \overline{S[q^{[1]}(z_k)]} \rangle < \langle \overline{S[q^{[1/2]}(z_k)]} \rangle. \tag{36}$$

The theoretical origin of this relation between the average KL distance and the average (maximized) entropy is not yet clear. Here is a concrete numerical example that illustrates (35, 36). For parameters of Table 2 we noted for $M = 55$: $\langle \overline{K}_1 \rangle = 1.756 < \langle \overline{K}_{1/2} \rangle = 1.758$ and $\langle \overline{S[q^{[1]}(z_k)]} \rangle = 4.006 < \langle \overline{S[q^{[1/2]}(z_k)]} \rangle = 4.008$.

