# OpenReview forum: "Maximum Entropy competes with Maximum Likelihood"
_ICLR.cc/2021/Conference — Reject_

### Official Review · AnonReviewer2 · 2020-10-25
**Maximum Entropy Competes with Maximum Likelihood.**

**Rating:** 6
**Confidence:** 4

**Review:**

Strengths:
1.	The paper is basically clear. The claims and method are correct.
2.	This paper proposes a novel MAXENT method to compete with maximum likelihood in sparse data. The topic of the paper is highly relevant to the ICLR community. The idea is original, and the motivation is strong in theoretical analysis. The theoretical analysis is presented in a fine manner. The numerical experiments show the validity of the proposed method.
3.    By assuming a well-defined prior Dirichlet density for unknown probabilities, it employs the average KL distance in evaluating the relevance of various MAXENT constraints, checking its general applicability, and comparing MAXENT with estimators having various degrees of dependence on the prior, the regularized ML and the Bayesian estimators.

Weakness:
1.	I’m confused about the application scenarios of the proposed method. The authors did not show any application value, though they claimed that MAXENT has a large number of applications in applied machine learning. The proposed method is not easy to understand and implement. The author should at least add one real data application.
2.	Differences between the proposed method and the existing methods (Good, 1970; Zhu et al,. 1997; Pandey & Dukkipati, 2013) should be stressed more. The differences can be presented theoretically and numerically.

---

> ### Author Response · Authors · 2020-11-23
> **Response to Reviewer 2**
>
>
> We thank Reviewer 2 for reporting on our paper and for making very useful suggestions.
>
> 1. Reviewer: I’m confused about the application scenarios of the proposed method. The authors did not show any application value, though they claimed that MAXENT has a large number of applications in applied machine learning. The proposed method is not easy to understand and implement. The author should at least add one real data application.
>
> Response: This is a good suggestion. We added Appendix A, where we explain in detail a real data application. We purposefully choose a difficult example for MAXENT, i.e. an example that contain categorical variables. This allows us to elucidate future research directions in this problem.
>
> 2. Reviewer: Differences between the proposed method and the existing methods (Good, 1970; Zhu et al,. 1997; Pandey & Dukkipati, 2013) should be stressed more. The differences can be presented theoretically and numerically.
>
> Response: Thank you for this suggestion. We added Appendix B, where we explain relations of our results with the minimum entropy proposed in (Good, 1970; Christensen, 1985; Zhu et al., 1997) and motivated via the minimum description length in (Pandey & Dukkipati, 2013). We are not yet able to state analytical results there, but numerical results are very suggestive.

---

### Official Review · AnonReviewer1 · 2020-10-28
**Too small numerical experiments**

**Rating:** 3
**Confidence:** 4

**Review:**

This paper considers the maximum entropy (MAXENT) method for estimating underlying probabilities over a finite alphabet, i.e., the multinomial model. The authors compare MAXENT with the regularized maximum likelihood, that is the Bayesian estimator under the Dirichlet prior with a common hyperparameter, and the Bayesian estimator with a general Dirichlet prior in terms of the Bayes risk, i.e., the KL-divergence from the true distribution to the estimated distribution averaged over the prior. The authors also consider the case where the prior is extended to the mixture of Dirichlet distributions. These comparisons are done numerically with synthetic data.

Although the practical performance of MAXENT is of interest, the paper provides little novel knowledge about it.

- The numerical experiments are at a too small scale. Without theoretical results, the experiments are required to be more comprehensive. The reported experiments are small scale both in terms of the alphabet size n and the data sequence length M.

- Isn’t it possible to include any discussion on hyperparameter estimation, for example, empirical Bayes since the hyperparameter estimation of the Dirichlet prior has long been studied?

- p.7, last paragraph: I wonder why the random perturbation of the ordering of Z devastates MAXENT since the hyperparametes \alpha can also be permutated accordingly.

Minor:
Right after eq. (15): \alpha_k^{[1]},..., \alpha_k^{[L]} (The last one should be \alpha_k^{[L]}.)
p.7, l.12 from the bottom: be be --> be

---

> ### Author Response · Authors · 2020-11-23
> **Response to Reviewer 1**
>
>
> We thank Reviewer 1 for reporting on our paper.
>
> 1. Reviewer: Although the practical performance of MAXENT is of interest, the paper provides little novel knowledge about it.
>
> Response: Let us list our new results concerning the performance of MAXENT.
>
> (1) We propose a criterion for deciding whether MAXENT estimators apply. It is that MAXENT estimators provide a smaller KL average distance than the data-free (prior-based) guess $q_k=1/n$.
>
> (2) We show that MAXENT estimators apply to sparse data, but require a specific prior information. Here sparse means $M<n$, i.e. the sample length $M$ is smaller than the number of probabilities $n$ to be inferred. We explored two different scenarios of such prior information. First, the unknown probabilities generated by homogeneous Dirichlet density (5) are most probably deterministic; please see sections 5.1. Second, there are prior rank correlations between the random quantity and its probabilities. This seems to be the simplest prior information that makes MAXENT applicable and superior over the optimally regularized maximum-likelihood estimator; please see sections 5.2.
>
> (3) Our approach is capable of describing several phenomena that are relevant for applying and understanding estimators: overfitting (i.e. adding more noisy constraints leads to poorer inference), instability of optimal Bayesian parametric estimators with respect to variation of prior details, inapplicability of non-parametric MAXENT estimators to very short samples etc.
>
> (4) We pose and partially solve two pertinent problems [see also Appendices A and B in this context]: the relevance of the affine symmetry (24) for the performance of MAXENT, and the application of MAXENT to categorical data.
>
> 2. Reviewer: The numerical experiments are at a too small scale. Without theoretical results, the experiments are required to be more comprehensive. The reported experiments are small scale both in terms of the alphabet size n and the data sequence length M.
>
> Response: We emphasize that some of our important results are analytic. This fact is mentioned by Reviewer 2. For analytical results please see Eqs. (9, 11, 14, 18, 24).
>
> In the new version we explored the whole range of $\alpha$ for the homogeneous Dirichlet prior; please see the newly added Table II. We explored a wider range for both $n$ and $M$, but we report numerical results for representative values of $M$ and $n$. These values ($n=60$ and various values of $M$) illustrate all the interesting effects we were able to detect. Since we explained in detail our numerical methods, people interested in other values of $n$ and $M$ can make their own simulations.
>
> 3. Reviewer: Isn’t it possible to include any discussion on hyperparameter estimation, for example, empirical Bayes since the hyperparameter estimation of the Dirichlet prior has long been studied?
>
> Response: This is a good suggestion. This discussion is included after (7). In this context we provided several new references.
>
> 4. Reviewer: p.7, last paragraph: I wonder why the random perturbation of the ordering of Z devastates MAXENT since the hyperparametes \alpha can also be permutated accordingly.
>
> If we permute the hyperparameters together with the values of Z, then nothing will change. The purpose of our remark is to confirm that for the given prior mixture the ordering of Z is important.

---

### Official Review · AnonReviewer4 · 2020-10-28
**A limited comparison between MaxEnt and regularized Maximum Likelihood using a Bayesian preformance measure**

**Rating:** 4
**Confidence:** 3

**Review:**

# Summary

This paper investigates maximum entropy (MaxEnt) inference and compares it to a Bayesian estimator and regularized maximum likelihood for finite models. To assess the accuracy of the different estimators, the authors use the average KL-divergence between the ground truth and the estimator, where the average is computed over all datasets of a given size $M$ and all probabilistic models of size $n$ (generated by some prior, either a single Dirichlet or a mixture of Dirichlets). Using numerical experiments, the authors find that the performance of MaxEnt deteriorates for sparse data generated from uniform models. However, by exploiting knowledge about the order of probabilities, MaxEnt can outperform regularized maximum likelihood.

# Assessment

I have to admit that I had different expectations based on the title and found the paper of limited interest. After reading the paper, I'm a bit puzzled as to what I learned and how much the findings can be generalized. Therefore, I would rather reject the paper in its current version.

Let me try to detail some of the problems that I have with the paper:

* The whole setting seems arbitrary to me. The author call their setup a "Bayesian approach" because the correct prior is used to assess the quality of an estimator (when computing $\langle \overline K\rangle$). In my understanding, a Bayesian approach would try to make reasonable assumptions about the likelihood and prior, integrate out any uninteresting parameters, and look at the resulting posterior.

* Typically, MaxEnt is used to incorporate expectations that are assumed to be known exactly. The paper seems to use MaxEnt as a blackbox method in order to circumvent the need of defining a prior. To apply MaxEnt, the data are transformed to moments (by computing sample means), which are then used to constrain the model. It is clear that for small $M$ the estimated expectations obtained by computing sample means are not very reliable and therefore MaxEnt (as used by the authors) performs purely on sparse data.

* The numerical experiments are run for very specific choices of $n$, $\alpha_k$, $z_k$, etc. There is no motivation for these choices. It remains unclear to me if the results generalize to other choices.

* As the authors acknowledge themselves the approach is limited to finite models with $n$ states. What about continuous or categorical variables?

---

> ### Author Response · Authors · 2020-11-23
> **Response to Reviewer 4**
>
> We thank Reviewer 4 for reporting on our paper.
>
> 1. Reviewer: The whole setting seems arbitrary to me. The author call their setup a "Bayesian approach" because the correct prior is used to assess the quality of an estimator (when computing the average KL distance). In my understanding, a Bayesian approach would try to make reasonable assumptions about the likelihood and prior, integrate out any uninteresting parameters, and look at the resulting posterior.
>
> Response: The response to this remark is basically similar to our last response to Reviewer 3.
>
> We need to emphasize the difference between the proper Bayesian statistics and the Bayesian decision theory that is applied in this paper. We cited the book by Cox & Hinkley that devotes much space for discussing this difference (please see sections 10 and 11 of the book). In the proper Bayesian statistics one employs the data for going from the prior to the posterior, and then works out various features of the posterior. For example, one can employ the posterior mean that does contain averaging over the prior.
>
> In the Bayesian decision theory there are two additional tools: decisions (estimators) and the average distance. Introducing the average distance allows to compare different estimators with each other, and this is what we do in the paper. Given a specific form of the distance, one can look at the best estimator, i.e. the one that minimizes the average distance. We recall that for the KL distance the optimal estimator is the posterior mean. In the proper Bayesian statistics the posterior mean is just introduced as a characteristic of the posterior. In the Bayesian decision theory the posterior mean is deduced from minimizing the KL distance. If another distance is preferred, then the optimal estimator will change.
>
> In the revised version we discussed these issues in detail; please see the changes done before and after Eq. (4), and after Eq. (9).
>
> 2. Reviewer: Typically, MaxEnt is used to incorporate expectations that are assumed to be known exactly.
>
> Response: This is what happens in statistical physics, where MAXENT originated. In physics there are conservation laws that allow precise knowledge of some constraints (energy, momentum, magnetization etc). We are not aware of any general principles that would fix the constraints precisely in statistics and machine learning. Constraints come from data and they are generally noisy.
>
> 3. Reviewer: To apply MaxEnt, the data are transformed to moments (by computing sample means), which are then used to constrain the model. It is clear that for small M the estimated expectations obtained by computing sample means are not very reliable and therefore MaxEnt (as used by the authors) performs purely on sparse data.
>
> Response: This is our basic motivation for studying this problem: the MAXENT constraints are necessarily noisy and it was unclear whether and to which extent they apply to sparse data. Our approach allows to design a well-defined criterion for this applicability: the average distance of a MAXENT estimator should be smaller than the average distance for the data-free (random guess) estimator (given by Eq.(17)) that is based on the prior information only. Armed by this criterion, we are able to show that MAXENT applies to sparse data provided that there are specific forms of prior information.
>
> 4. Reviewer: The numerical experiments are run for very specific choices of $n$, $\alpha_k$ , $z_k$ , etc. There is no motivation for these choices. It remains unclear to me if the results generalize to other choices.
>
> Response: In the present version we explored the whole range of $\alpha$ (from $\alpha$ close to zero to a sufficiently large value of $\alpha$); please see sections 5.1 and 5.2.
>
> We did experiments for various values of $n$ and $M$ and noted that our choice of $n=60$ is sufficiently representative, i.e. no new effects appear for other choices of $n$. As for the choice of $z_k=k$ that was employed in numerical experiments, we note that our results do not change upon using any other choice of $z_k$ that is related to $z_k=k$ by means of the affine symmetry (24). In fact, we tried other choices of $z_k$ (e.g. the random choice of them), but the results were mostly worse than for the choice $z_k=k$. In that sense the choice $z_k=k$  is a reasonable one for theoretical studies.
>
> 5. Reviewer: As the authors acknowledge themselves the approach is limited to finite models with states. What about continuous or categorical variables?
>
> Response: For continuous variables we expect mostly technical difficulties, e.g. the maximum entropy should be changed to the minimum relative entropy, Dirichlet density should be changed to a reasonable parametric density etc. The situation with categorical variables is more challenging. In the newly added Appendix A we study this question in a real data example and show that MAXENT does apply. These encouraging preliminary results need further development.

---

### Official Review · AnonReviewer3 · 2020-10-28
**Average KL does not seem to be correct**

**Rating:** 4
**Confidence:** 5

**Review:**

In this paper, the authors discussed the maximum entropy method of obtaining estimators for discrete probability distribution. They also gave an exposition regarding different moment type constraints and how they behave under affine data transformation. They authors then performed numerical experiments comparing the maximum entropy method under different constraints and the regularized maximum likelihood estimator.

I find that overall the writing is fine but some parts were quite difficult to understand. For example, I found it hard to follow the discussions concerning equations (20) to (24). What do you mean by fixing the second moment? Are you setting (23) to a constant?

However I think the main problem of this paper is the use of average KL to measure performance given in (5). The use of (5) as a lost function does not seem to be correct. True $q$ is a fixed quantity and you cannot directly put a prior on true parameter values. It is easy to be confused when you are talking about Bayesian procedures and true parameter values. So let me try to break this down. Let us assume that the data is generated from some unknown true distribution $Z_{i_1},\dotsc,Z_{i_M}\sim P(Z=z_k)=q_k^0$ for $k=1,\dotsc,n$, where $q_k^0$ is unknown. Since we do not know the true distribution, we come up with a model for the observed data $Z_{i_1},\dotsc,Z_{i_M}$, say $Z_{i_1},\dotsc,Z_{i_M}\sim P(Z=z_k)=q_k$. So the $q_k$'s here are the parameters for our model which we need to estimate. Let us do the Bayesian way by putting a prior on the $q_k$'s, for example $(q_1,\dotsc,q_n)\sim\mathrm{Dirichlet}(\alpha_1,\dotsc,\alpha_n)$. For there we can get the posterior $\Pi(q_1,\dotsc,q_n | Z_{i_1},\dotsc,Z_{i_M})$ by Bayes theorem and the posterior mean $\hat{q}_k=\mathrm{E}(q_k|Z_{i_1},\dotsc,Z_{i_M})$. So the $hat{q}_k$ is supposed to be the estimator for the unknown $q_k^0$ and we can quantify their distance using KL by
$$K(q^0,\widehat{q})=\sum_{k=1}^nq_k^0\log\frac{q_k^0}{\widehat{q}_k}.$$

In the paper, the authors endowed a Dirictlet prior on the $q_k^0$'s, and this is not correct. In Bayesian statistics, you endow a prior on the model of the truth and not directly on the truth. From here you can talk about things like how does my estimator performed across all different kinds of truth, i.e.,
$$
\sup_{q^0\in\mathcal{S}}K(q^0,\widehat{q})
$$
where $\mathcal{S}$ is the simplex in $\mathrm{R}^n$.

Typos:
1. Page 1, 2nd line from below, main results are that...

2. The line after (15), do you mean $\alpha_k^{[1]},\dotsc,\alpha_k^{[L]}$? Then the line after this,
Below we show some concrete...

3. The 3rd line after (24), (17) is a concave function in $\{q_k\}_{k=1}^n$

4. In (7), the range of integration is over the simplex $\{(q_1,\dotsc,q_n):\sum_{k=1}^np_k=1,0\leq p_k\leq1,\forall k\}$.

---

> ### Author Response · Authors · 2020-11-23
> **Response to Reviewer 3**
>
> We thank Reviewer 3 for reporting on our paper.
>
> Reviewer 3 has several minor remarks and one basic criticism. We start with minor ones.
>
> 1. Reviewer: I found it hard to follow the discussions concerning equations (20) to (24).
>
> Response: The whole this part was rewritten.
>
> 2. Reviewer: What do you mean by fixing the second moment? Are you setting (23) to a constant?
>
> Response: Yes, we set it to a constant during the entropy maximization. This constant equals the second empiric moment deduced from the sample (1).
>
> We corrected all typos noted by Reviewer 3.
>
> The basic criticism of Reviewer:
>
> 3. Reviewer: However I think the main problem of this paper is the use of average KL to measure performance given in (5). The use of (5) as a lost function does not seem to be correct. True q is a fixed quantity and you cannot directly put a prior on true parameter values. ….. In the paper, the authors endowed a Dirictlet prior on the $q_k^0$'s, and this is not correct. In Bayesian statistics, you endow a prior on the model of the truth and not directly on the truth.
>
> Response: First of all, we should emphasize the difference between the proper Bayesian statistics and the Bayesian decision theory that is applied in this paper. We cited the book by Cox & Hinkley that devotes much space in discussing this difference (please see sections 10 and 11 of the book). In the proper Bayesian statistics one employs the data for going from the prior to the posterior, and then works out various features of the posterior. For example, one can employ the posterior mean that does contain averaging over the prior.
>
> In the Bayesian decision theory there are two additional tools: decisions (which in our case correspond to estimators) and the average utility that in our situation refers to the average distance. Importantly, the average is taken over both the data and prior. Introducing the average distance allows to compare different estimators with each other, and this is what we do in the paper. Given a specific form of the distance, one can look at the best estimator, i.e. the one that minimizes the average distance. This is a well posed question, and in our paper we recall that for the KL distance the optimal estimator is just the posterior mean. Please note the difference here: in the proper Bayesian statistics the posterior mean is just introduced as a characteristic of the posterior. There are other characteristics and it is strictly speaking unclear why we should stick to the posterior mean. In the Bayesian decision theory the posterior mean is deduced from minimizing the KL distance. If another distance is preferred, then the optimal estimator is not anymore the posterior mean.
>
> We agree that in the first version of our paper we did not spend enough time on discussing the above differences, but this drawback is improved in the new version. Please see the changes done before and after Eq. (4), and after Eq. (9). In particular, we emphasize that we are doing the standard Bayesian decision theory and contrast it with the proper Bayesian statistics.
>
> Reviewer 3 emphasizes that unknown probabilities are to be considered as fixed numbers, despite of the fact that the usage of the posterior mean is allowed. We do not understand why Reviewer 3 believes that the average KL-distance necessarily assumes that unknown probabilities are not anymore fixed numbers. Indeed, the average KL distance involves the averaging over data samples that is obviously compatible with the fixedness of unknown probabilities. The average KL distance also involves the averaging over the prior density, but the same average is involved in the posterior mean, which is admitted by Reviewer 3. Hence we do not see a contradiction between what Reviewer 3 advocates and what we are doing.
>
> We close by emphasizing again that we are doing the standard Bayesian decision theory that was/is applied for deciding on the usage of various point estimators. The argument by Reviewer 3 does not show any contradiction in our approach.

---

### Author Response · Authors · 2020-11-23
**List of changes in the revised paper**

We are grateful to all Reviewers for reporting on our manuscript. Detailed responses are given to each Reviewer separately. Here we explain changes made in the revised version of the paper.

1. The abstract was rewritten.

2. The difference between the proper Bayesian statistics and the Bayesian decision theory is underlined at several places  (answers critical remarks by Reviewer 3 and Reviewer 4).

3. A new section 5.1 was added that does extend the range of our numeric results (answers critical remarks by Reviewer 1 and Reviewer 4).

4. A new Table II was added.

5. Appendix A was added following the suggestions by Reviewer 2 and Reviewer 4. It explains the applicability of our results to a real categorical data.

6. Appendix B was added following the suggestion by Reviewer 2. In particular, it explains relations of our results to the minimum entropy principle.

7. Section 4 was rewritten (partially based on the suggestion by Reviewer 3).

8. New references on hyper-parameter estimation were added following the suggestion by Reviewer 1.

9. Notations were changed to make derivations clearer.

10. Several pertinent references were added.

---

### Decision · Program_Chairs · 2021-01-07
**Final Decision**

**Decision:**

Reject

**Comment:**

As one of the reviewers concisely summarized: This paper investigates maximum entropy (MaxEnt) inference and compares it to a Bayesian estimator and regularized maximum likelihood for finite models.

Two reviewers specifically question whether they have learned anything new after reading. This combined with various other drawbacks described during the review phase led to strong agreement among the reviewers about a variety of deficiencies in this paper. One reviewer initially gave a relatively high score but has since revised his/her opinion in light of the other reviews and discussion. I find that the significance of this work is not high enough to warrant acceptance at this time, but the authors would do well to incorporate the reviewers suggestions to improve the paper.